# Histone H3.3 promotes IgV gene diversification by enhancing formation of AID-accessible single-stranded DNA

Marina Romanello[†], Davide Schiavone[†], Alexander Frey & Julian E Sale[*]

## Abstract

Immunoglobulin diversification is driven by activation-induced deaminase (AID), which converts cytidine to uracil within the Ig variable (IgV) regions. Central to the recruitment of AID to the IgV genes are factors that regulate the generation of single-stranded DNA (ssDNA), the enzymatic substrate of AID. Here, we report that chicken DT40 cells lacking variant histone H3.3 exhibit reduced IgV sequence diversification. We show that this results from impairment of the ability of AID to access the IgV genes due to reduced formation of ssDNA during IgV transcription. Loss of H3.3 also diminishes IgV R-loop formation. However, reducing IgV R-loops by RNase HI overexpression in wild-type cells does not affect IgV diversification, showing that these structures are not necessary intermediates for AID access. Importantly, the reduction in the formation of AID-accessible ssDNA in cells lacking H3.3 is independent of any effect on the level of transcription or the kinetics of RNAPII elongation, suggesting the presence of H3.3 in the nucleosomes of the IgV genes increases the chances of the IgV DNA becoming single-stranded, thereby creating an effective AID substrate.

**Keywords** activation-induced deaminase (AID); histone H3.3; immunoglobulin diversification; single-stranded DNA; transcription

**Subject Categories** Chromatin, Epigenetics, Genomics & Functional Genomics; DNA Replication, Repair & Recombination; Immunology

The EMBO Journal (2016) 35: 1452–1464

## Introduction

The diversification of immunoglobulin genes by activation-induced deaminase (AID) is crucial for affinity maturation and effective adaptive immunity (Muramatsu *et al*, 2000; Revy *et al*, 2000). AID deaminates cytidine (dC) to uracil (dU) within the Ig variable (IgV) regions, triggering somatic hypermutation and, in some species including birds, gene conversion, a process by which the rearranged V gene recombines with upstream pseudogenes (Di Noia & Neuberger, 2007).

Activation-induced deaminase acts only on single-stranded DNA (ssDNA) (Bransteitter *et al*, 2003; Dickerson *et al*, 2003), and therefore, *in vivo*, IgV deamination requires the exposure of this substrate. Multiple lines of evidence have linked the generation of ssDNA and the recruitment of AID to the level of transcription (Peters & Storb, 1996; Fukita *et al*, 1998; Ramiro *et al*, 2003). AID activity at the IgV genes is thus associated with chromatin features characteristic of actively transcribed loci including DNA hypomethylation, enrichment for the histone modifications H3K4me3 and H3K36me3 and the variant histone H3.3 (Fraenkel *et al*, 2007; Begum *et al*, 2012; Aida *et al*, 2013). However, whether any of these features promotes the activity of AID directly, that is independently of simply promoting transcription, remains unclear. While early work on this question demonstrated that the distribution of "active" histone modifications, including H3K4me3 and H3/H4 acetylation, does not accurately delineate the zone of mutation in the Ig loci (Odegard *et al*, 2005), a more recent correlative genome-wide study linked H3K27ac and H3K36me3, modifications characteristic of enhancers and transcribed genes, respectively, to AID activity at off-target sites (Wang *et al*, 2014). H3.3 has also been proposed as a "target marker" for somatic hypermutation (Aida *et al*, 2013). However, while H3.3 is enriched in the chromatin of loci that support hypermutation, a causal link between H3.3 and the level of diversification has not been established.

The function of H3.3 remains incompletely understood. While H3.3 differs from canonical vertebrate H3 by only four amino acids, it binds distinct histone chaperones (Szenker *et al*, 2011) and nucleosomes containing H3.3 are unusually sensitive to salt-dependent disruption (Jin & Felsenfeld, 2007). Importantly, H3.3 is expressed throughout the cell cycle and can therefore be incorporated independently of S-phase (Tagami *et al*, 2004). It is thus enriched in areas of increased nucleosome turnover including transcribed genes but is also found in silent regions including pericentromeric heterochromatin and telomeres (Szenker *et al*, 2011). H3.3 is essential for development as it is necessary for proper trimethylation of H3K27 at genes regulated by the polycomb-repressive complex 2 (Banaszynski *et al*, 2013). Complete absence of H3.3 in mice thus results in peri-implantation lethality (Jang *et al*, 2015).

Medical Research Council Laboratory of Molecular Biology, Cambridge, UK
*Corresponding author. Tel: +44 1223 267099; E-mail: jes@mrc-lmb.cam.ac.uk
†These authors contributed equally to this work

 

We recently reported a mutant chicken DT40 cell line that completely lacks H3.3 (hereafter termed *h3.3* cells; Frey *et al*, 2014). Despite its association with actively transcribed genes (Ahmad & Henikoff, 2002) and its role in controlling expression from developmentally regulated loci (Banaszynski *et al*, 2013), H3.3 does not appear to be necessary to maintain active transcription of the majority of genes in DT40 (Frey *et al*, 2014), which is derived from a differentiated B cell (Baba *et al*, 1985).

The Ig loci of DT40 are continuously diversified by a combination of gene conversion and non-templated point mutation, closely mimicking the *in vivo* behaviour of avian B cells (Buerstedde *et al*, 1990; Kim *et al*, 1990). *h3.3* DT40 cells thus provide an ideal system in which to examine the effect loss of H3.3 has on IgV gene diversification. In this paper, we show that the absence of H3.3 significantly impairs diversification of the rearranged Ig light chain locus (*IGVL*$_R$) of DT40 independently of any demonstrable effect on its transcription. Rather, we find that H3.3 influences the formation of AID-accessible ssDNA during Ig gene transcription, providing clear evidence that nucleosome composition can directly influence Ig diversification by controlling the formation of the single-stranded DNA substrate of AID.

# Results

### Diminished diversification of the rearranged Ig light chain locus (*IGVL*$_R$) in DT40 cells lacking H3.3

To examine Ig diversification in our *h3.3* mutant of DT40 (Frey *et al*, 2014), we initially sequenced randomly selected V genes from subclones of two independent *h3.3* lines that had been expanded for 36 divisions (Frey *et al*, 2014). This revealed a striking decrease in the frequency of both gene conversion and non-templated point mutation in the absence of H3.3 (Fig 1A and B) that was not attributable to loss of endogenous AID expression (Fig 1C). The constitutive Ig gene diversification in DT40 cells results in the formation of variants that have lost surface IgM expression either due to nonsense mutation or mutations that impair heavy and light chain pairing (Sale *et al*, 2001). The formation of these surface Ig (sIg) loss variants in multiple parallel clones grown for a known number of cell divisions can be used to monitor the rate of Ig diversification by fluctuation analysis (Sale, 2012). Diversification in the absence of H3.3 resulted in the generation of fewer loss variants (Fig 1D). However, the small signal-to-noise ratio of this assay in the gene converting DT40 with natural AID expression led us to examine whether this defect was also observed over a range of higher AID activity levels. We thus created clones overexpressing an enzymatically hyperactive form of AID, hAID$^{up}$ 7.3 (hereinafter referred to as hAID$^{up}$; Wang *et al*, 2009). This AID variant exhibits increased levels of deamination, but generates an identical pattern of deamination to the wild-type enzyme (Wang *et al*, 2009; Taylor *et al*, 2014). hAID$^{up}$ expression resulted in robust generation of sIg-loss variants in wild-type cells (Fig 1D). However, when expressed in *h3.3* cells, it results in significantly lower rates of sIg-loss variant production (Fig 1D). This defect was reversed by ectopic expression of H3.3. We next confirmed the reduction in overall Ig diversification in *h3.3* cells by direct sequencing of V gene libraries from expanded clones (Fig 1E). To assess whether the reduction in diversification in *h3.3*

cells is consistent over a range of deamination levels, we correlated hAID$^{up}$ cDNA levels with the rate of diversification, monitored as the number of mutations acquired per base pair per generation, in multiple clones (Fig 1F). This revealed that for any given level of hAID$^{up}$ expression, diversification in *h3.3* cells was reduced relative to wild type (Fig 1F). However, there was no change in the pattern of diversification. Both the balance between gene conversion and non-templated point mutation (Fig 1G) and the spectrum of point mutations were unaltered (Fig EV1), suggesting a deficiency in the entire process of *IGVL*$_R$ diversification.

### The reduction in Ig diversification in *h3.3* cells reflects diminished AID activity at *IGVL*$_R$

We considered two possible explanations for the reduced Ig diversification in *h3.3* cells. Either there was less deamination of the *IGVL*$_R$ target, or the resulting abasic sites were less likely to be processed through a mutagenic pathway (Fig 2A). To discriminate between these two possibilities, we inhibited the conversion of dU to abasic sites by expression of a bacteriophage-derived uracil glycosylase (UNG) inhibitor (UGI) (Di Noia & Neuberger, 2002). Inhibition of UNG leads to direct replication of the unexcised uracils resulting exclusively in C-to-T transitions, effectively revealing the "footprint" of deamination by AID (Xue *et al*, 2006). We anticipated that if the reduced Ig diversification in *h3.3* cells reflected diminished AID activity, this would result in reduced AID "footprint" mutations in the presence of UGI. If, however, the defect resulted from normal AID activity with reduced mutagenic processing of dU, we would observe similar levels of C-to-T transitions in both wild-type and *h3.3* cells. Following UGI expression, we observed that Ig mutagenesis comprised almost exclusively C-to-T transitions, as expected (Figs 2B and EV1). However, there remained far fewer mutations in *h3.3* cells than in wild type (Fig 2B), a difference that was retained over a range of AID$^{up}$ expression levels (Fig 2C). Thus, the diminished Ig diversification in *h3.3* cells is due to impaired activity of AID on the IgV locus, rather than any reduction in the downstream mutagenic processing of dU. These results show that H3.3 is necessary for efficient AID activity on the IgV locus.

### H3.3 is enriched in *IGVL*$_R$ but is not required for normal transcriptional activity of the locus

Enrichment of H3.3 has been linked with transcriptional activity and indeed has been proposed to contribute to ensuring that full transcription is maintained through cell division (Ahmad & Henikoff, 2002). However, we recently showed that surprisingly few genes exhibit significantly dysregulated expression in DT40 cells lacking H3.3 (Frey *et al*, 2014). Since Ig diversification is intimately linked to transcriptional activity, we carefully examined expression of *IGVL*$_R$ in *h3.3* cells. As expected, chromatin immunoprecipitation revealed that H3.3 is enriched in the rearranged and diversifying *IGVL*$_R$ allele compared with the untranscribed and mutationally silent (Sale *et al*, 2001) unrearranged *IGVL*$_{UR}$ allele (Fig 3A). However, removing H3.3 did not affect the steady-state levels of IgVλ mRNA (Fig 3B) or surface IgM protein (Fig 3C). Moreover, the enrichment of H3K4me3 and H3K36me3, histone modifications associated with the TSS of genes and RNA polymerase II (RNAPII) elongation, respectively, was identical in wild-type and *h3.3* cells

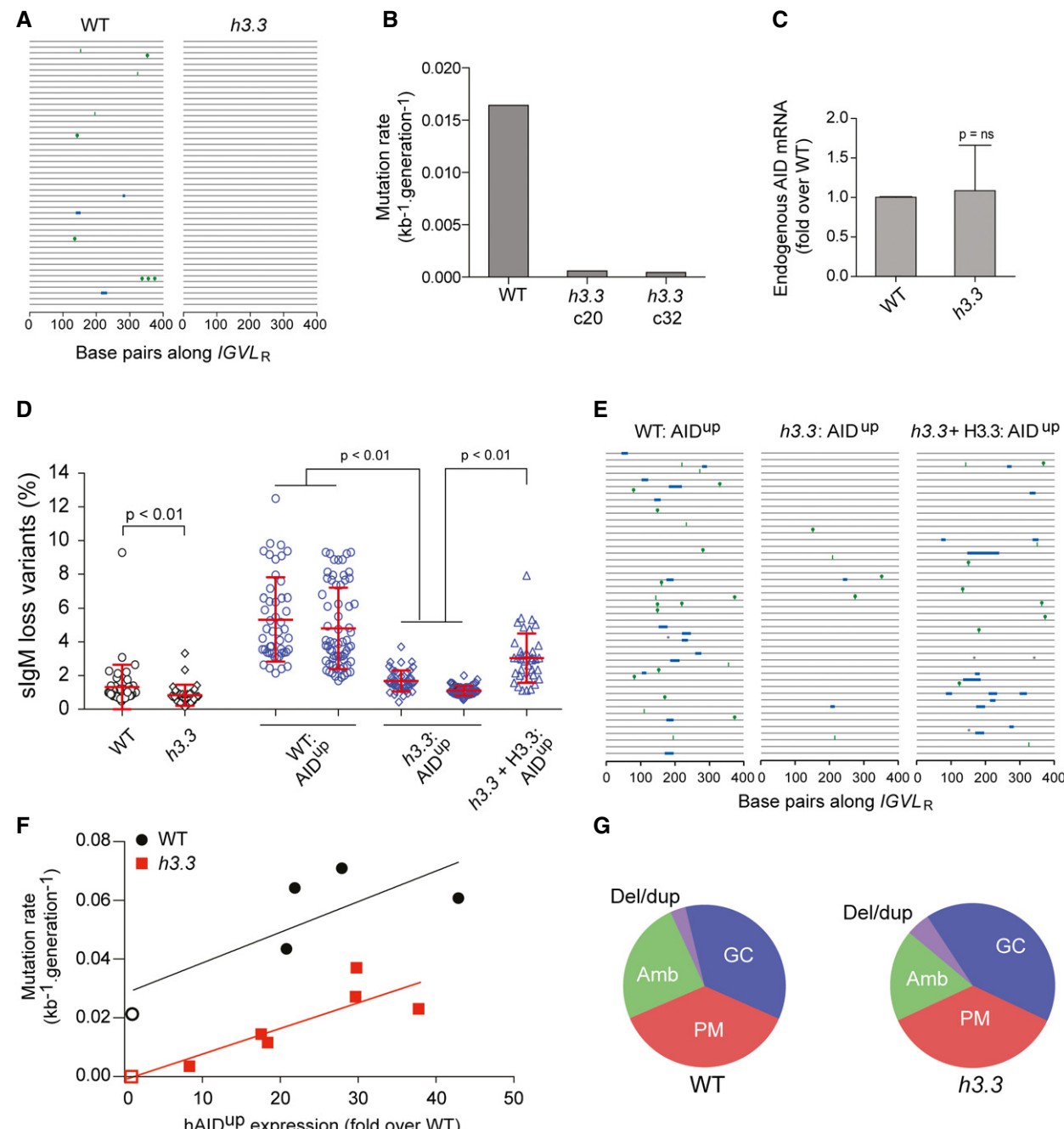

**Figure 1.   Reduced diversification of *IGVL* in cells lacking H3.3.**

A   Ig sequence diversification in wild type and *h3.3*. Each line represents an *IGVL*R gene from a population cells that have expanded for 36 divisions after cloning. Blue bars = gene conversions; green lollipops = point mutations; green bars = ambiguous mutations (Sale *et al*, 2001).

B   Quantification of sequence diversification in wild-type and two *h3.3* clones expressed as unique events (gene conversions or point mutations) per kb per generation. Sequences analysed: WT $n = 142$, for *h3.3* c20 $n = 140$, for *h3.3* c32 $n = 190$.

C   Quantification of endogenous AID mRNA in wild-type and *h3.3* cells. The graph represents the mean and SD of four independent RNA extractions of wild type and *h3.3c20*. *P*-value calculated with the unpaired *t*-test.

D   Fluctuation analysis for the formation of sIg-loss variants in wild-type (black circles) and *h3.3* clones (black diamonds) and in two wild-type clones (blue circles), two *h3.3* clones (blue diamonds) expressing hAID[up] (Wang *et al*, 2009) at 15–20 times the level of wild-type AID. The final column shows *h3.3* cells with hAID[up] complemented with H3.3 (blue triangles). Clones were expanded for 36 cell divisions at which point the percentage of sIgM-negative cells in each population was determined by flow cytometry. Error bars represent mean ± SD. *P*-values calculated with the Mann–Whitney *U*-test.

E   Ig sequence diversification in wild-type, *h3.3* and complemented *h3.3* cells overexpressing hAID[up]. Key as in (A) plus * = deletion or duplication.

F   Mutation rate as a function of hAID[up] expression in wild type and *h3.3*. Filled symbols represent clones overexpressing hAID[up]; open symbols represent clones with natural AID expression. Linear regression lines are shown.

G   Spectrum of diversification in wild type and *h3.3*. GC = gene conversion; PM = point mutation; Amb = ambiguous; Other = deletions or duplications.

(Fig 3D). Together, these observations show that the level of transcription of $IGVL_R$ is not altered by loss of H3.3. Thus, the presence of H3.3 in nucleosomes influences Ig diversification independently of any effect on the level of transcription.

Pausing or stalling of RNAPII has been proposed to be key to generating the ssDNA necessary for AID action (Kenter, 2012; Keim

et al, 2013), so we next asked whether, despite the evidence for steady-state transcription being unaffected, H3.3 modulates the passage of RNAPII through $IGVL_R$. To assess this, we measured transcription elongation rates by arresting RNAPII at the TSS with 5,6-dichloro-1-β-D-ribofuranosylbenzimidazole (DRB), and then monitoring the incorporation of 4-thiouridine (4SU) into nascent mRNA on release of the arrested RNAPII following washout of the DRB (Fuchs *et al*, 2015). The kinetics of 4SU incorporation across $IGVL_R$, and a control locus *GAPDH*, was indistinguishable between wild-type and *h3.3* cells (Figs 3E and EV2). Thus, H3.3 does not detectably influence RNAPII elongation, suggesting that a significant change in pausing of the polymerase is unlikely to explain the observed differences in the recruitment of AID to $IGVL_R$ seen in *h3.3* cells.

## AID localisation and enzymatic activity are unaffected by loss of H3.3

We considered a number of further possible explanations for the reduced activity of AID at $IGVL_R$ in *h3.3* cells. Although only a relatively small number of loci exhibit dysregulated transcription in *h3.3* DT40, we examined our previously published expression data (Frey *et al*, 2014) for reported interactors or regulators of AID. None exhibited significantly different levels of expression between wild-type and *h3.3* cells (Table EV1). A direct interaction between AID and H3.3 has not been reported, and neither have we detected any evidence of such an interaction in pull-downs interrogated by Western blotting or by mass spectrometry (data not shown). We were also unable to detect any gross changes in the subcellular distribution of AID in *h3.3* cells (Fig EV3). We next considered the idea that AID activation is driven by the DNA damage response (Vuong *et al*, 2013), especially in the light of the DNA damage sensitivity of *h3.3* cells (Frey *et al*, 2014). However, the S-phase checkpoint response monitored by phosphorylation of CHK1 and H2Ax following exposure to UVC light was indistinguishable between wild-type and *h3.3* cells (Fig EV4), making it unlikely that a defect in DNA damage signalling in *h3.3* cells accounts for reduced Ig diversification. To assess directly whether AID from *h3.3* cells is less enzymatically active than that from wild type, we pulled down AID protein from total and nuclear extracts of both cell lines and carried out an *in vitro* deamination assay (Petersen-Mahrt & Neuberger, 2003). This revealed no difference in the kinetics of deamination by the AID isolated from either line (Fig EV5). Finally, we confirmed that the AID transgene itself was not mutated in *h3.3* cells (data not shown). Together, these results suggest that the impaired activity of

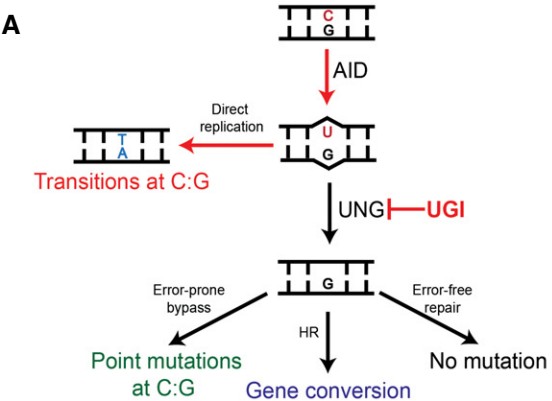

**A**

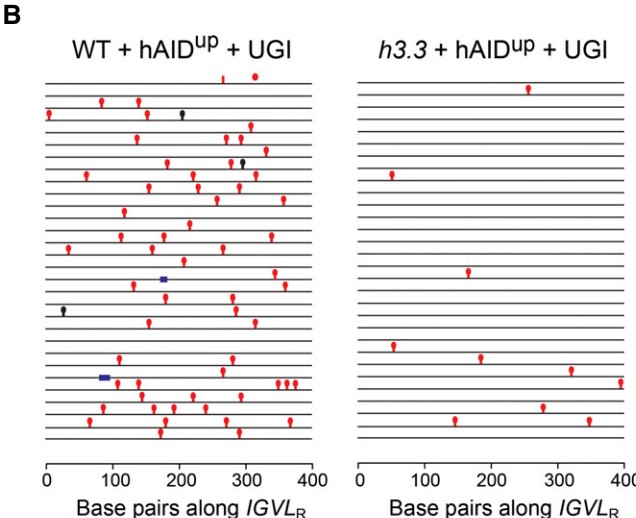

**B**

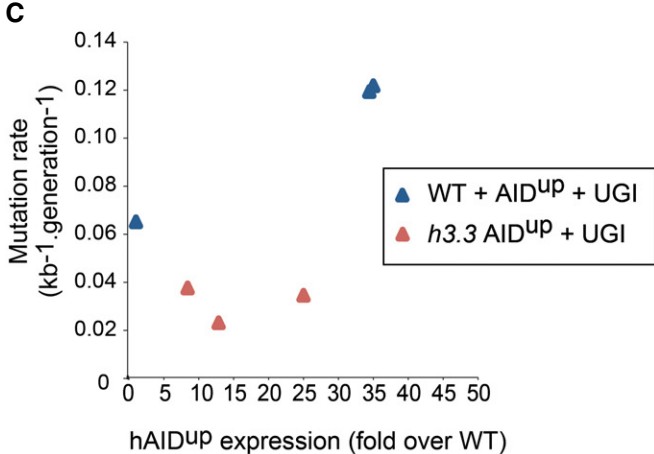

**C**

**Figure 2. The limited IgV diversification in *h3.3* cells results from reduced activity of AID.**

A  Scheme outlining the mechanism of Ig diversification in DT40 cells, indicating the action of uracil glycosylase inhibitor (UGI). UNG = uracil DNA glycosylase.

B  Diversification in wild-type and *h3.3* cells expressing similar levels of hAID[up] and UGI. Red lollipops = transitions at C/T; black lollipops = other point mutations; blue bars = gene conversions.

C  Mutation rate as a function of hAID[up] expression in UGI-expressing wild-type and *h3.3* cells. Each clone was expanded for 36 cell divisions and hAID[up] expression determined by qPCR during the last week of culture.

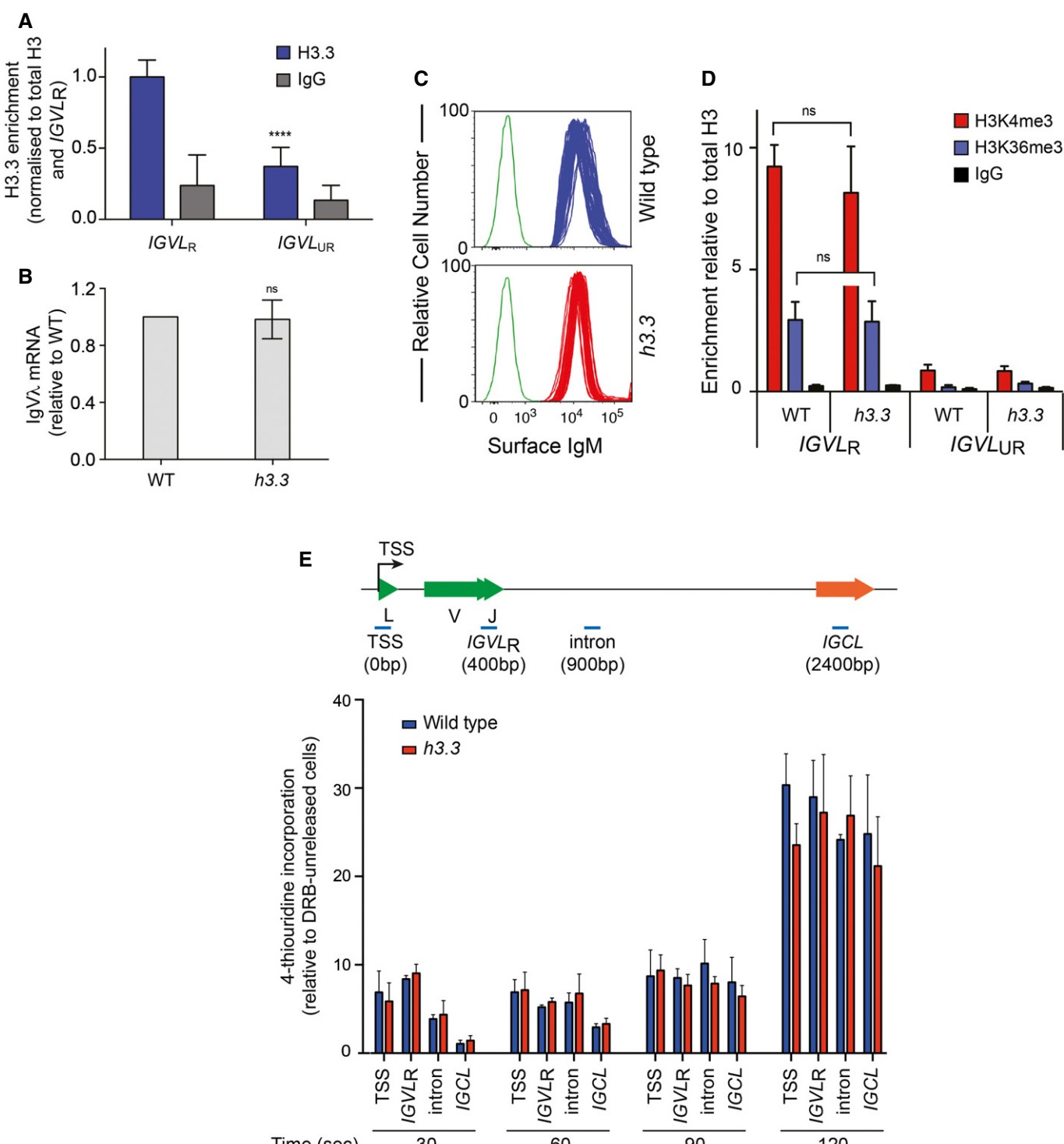

**Figure 3.  Loss of H3.3 does not affect transcription of _IGVL_.**

A    Enrichment of H3.3 in the rearranged allele of the Ig light chain ($IGVL_R$) compared with the unrearranged allele ($IGVL_{UR}$). Each bar represents the mean of 3 independent ChIPs each assayed with 3 repeat qPCRs (****$P < 0.0001$, unpaired $t$-test).

B    Igλ mRNA levels in wild-type and _h3.3_ cells determined by qPCR, normalised to EF-1α expression and to WT. $n = 3$ (ns = not significant, unpaired $t$-test).

C    Surface Ig (sIg) in multiple clones of wild-type (blue) and _h3.3_ (red) cells. Green = unstained control.

D    ChIP for H3K4me3 and H3K36me3 in $IGVL_R$ and $IGVL_{UR}$. Each bar represents the mean of 3 ChIPs for H3K4me3 and IgG and 2 for H3K36me3, each assayed with 3 qPCR repeats (statistical analysis as in panel A).

E    4sUDRB mapping of transcription kinetics through _IGVL_. Representative locations of qPCR amplicons (blue) are shown relative to the TSS of _IGVL_ (distance is shown in parenthesis). qPCR signals were normalised to the enrichment at time zero (before release). Error bars represent ± SD of qPCR triplicates.

AID on the $IGVL_R$ locus is not linked to defective AID localisation or enzymatic activity.

### IGVL R-loop formation is reduced in the absence of H3.3

Given that the intrinsic activity of AID is unaffected by the absence of H3.3, we next asked whether the formation of the substrate of AID, ssDNA, was reduced in *h3.3* cells. ssDNA in transcribed regions can form R-loops, DNA–RNA hybrids between nascent mRNA and single-stranded regions on the DNA template (Santos-Pereira & Aguilera, 2015). R-loops levels thus provide an indirect indication of the single strandedness of a locus. It has also been proposed that R-loops may facilitate Ig diversification: although AID is unable to access DNA when in a DNA–RNA hybrid (Bransteitter *et al*, 2003), R-loops may help to stabilise ssDNA allowing AID access to the opposite strand (Gómez-González & Aguilera, 2007). To assess the formation of R-loops in $IGVL_R$, we employed DNA–RNA hybrid immunoprecipitation (DRIP) with the S9.6 antibody (Boguslawski *et al*, 1986). DRIP of $IGVL_R$ in wild-type cells revealed significant enrichment of R-loops in the variable region (*IGVL*) compared with the constant region (*IGCL*; Fig 4A). However, in *h3.3* cells the DRIP signal in $IGVL_R$ was significantly reduced (Fig 4A). *In vitro* RNase H treatment of the samples prior to immunoprecipitation with S9.6 reduced the signal in both cases to nearly background levels (Fig 4A, RNase H treatment), confirming the specificity of the assay for the detection of R-loop. Thus, IgV R-loop levels correlate with the level of diversification.

### H3.3 does not influence overall nucleosome positioning in *IGVL*

H3.3 has been proposed to influence nucleosome positioning (Thakar *et al*, 2009; Le *et al*, 2010), which could in turn affect the formation of single-stranded DNA. Further, strong nucleosome positioning sequences have been shown to impair diversification (Kodgire *et al*, 2012). We therefore asked whether the reduced R-loop formation in *h3.3* cells reflected a change in the distribution of nucleosomes across the locus by mapping the average nucleosome positions in *IGVL* in wild-type and *h3.3* cells using micrococcal nuclease treatment coupled to quantitative PCR (Singh *et al*, 2013). This revealed two centres of nucleosome occupancy in *IGVL* of wild-type cells (Fig 4B). However, the average position of these nucleosomes was the same in both wild-type and *h3.3* cells, suggesting that a change in nucleosome distribution is not the explanation for the reduction in *IGVL* R-loop formation or Ig diversification (Fig 4B).

### R-loop formation is not necessary for efficient access of AID to *IGVL*

The correlation between *IGVL* R-loop levels and diversification does not prove that R-loops are required for diversification despite the attractive idea that they could stabilise ssDNA for AID access (Gómez-González & Aguilera, 2007). To address whether R-loop formation is necessary for Ig diversification in DT40, we expressed both chicken RNase HI and hAID[up] in wild-type DT40 cells (Fig EV6). DRIP analysis demonstrated that the expression of RNase HI reduced R-loop formation in *IGVL* to near background levels (Fig 4C). However, it did not affect Ig diversification (Fig 4D),

consistent with the observations of Martin and colleagues in hypermutating human B cells (Parsa *et al*, 2012). Thus, while H3.3 promotes the formation of R-loops in *IGVL,* R-loops are not necessary intermediates in AID-dependent Ig diversification. This suggested that H3.3 might be directly promoting the formation of single-stranded DNA.

### Reduced formation of short ssDNA tracts in *h3.3* cells

To test this idea, we employed *in situ* bisulphite mapping of $IGVL_R$ to directly monitor formation of single-stranded DNA (Ronai *et al*, 2007). Bisulphite deaminates dC present in single-stranded DNA at the time of treatment, which can be detected by subsequent sequencing, the resulting dU being read as dT. Using this approach, short patches of single-stranded DNA (ssDNA) have previously been reported on both strands of the Ig V genes in a hypermutating human B-cell line and primary murine B cells (Ronai *et al*, 2007; Parsa *et al*, 2012). Applying this technique to DT40 cells also revealed tracts of ssDNA throughout $IGVL_R$ on both strands (Fig 5A). The prevalence of these ssDNA tracts was significantly reduced when H3.3 was absent (Fig 5A and B). This reduction was reversed by ectopic expression of H3.3 (Fig 5B). Importantly, in contrast to *IGVL* R-loops, the formation of ssDNA detectable directly with bisulphite is not sensitive to RNase HI (Fig 5C), confirming that R-loops in *IGVL* are a consequence of ssDNA formation, not its cause. Together, our results demonstrate that H3.3 directly promotes the formation or stabilisation of AID-accessible ssDNA.

## Discussion

In this paper, we have demonstrated that H3.3 exerts a significant influence over the ability of AID to access and mutate the DNA of $IGVL_R$ in DT40 cells. H3.3 does this by enhancing the exposure of tracts of single-stranded DNA without affecting the overall kinetics or level of transcription of the locus.

Multiple lines of evidence have linked the mutation of IgV genes to their transcription (Peters & Storb, 1996; Goyenechea *et al*, 1997; Ramiro *et al*, 2003; Taylor *et al*, 2014). Since the enzymatic activity of AID requires the exposure of single-stranded DNA (Bransteitter *et al*, 2003; Dickerson *et al*, 2003), a key question is how transcription generates this substrate and why this process is sensitive to H3.3. AID is able to target the single-stranded DNA generated when RNAPII is loaded and poised for transcription at the start of genes in yeast (Taylor *et al*, 2014), suggesting that the ssDNA structures generated by the preinitiaition complex are accessible to AID even in the absence of B cell, or even vertebrate-specific, factors. However, mutations in IgV of B cells are found up to 2 kb from the TSS, far beyond the region occupied by the pre-initiation complex, and thus, additional mechanisms must be required to extend the mutation domain away from the site of RNAPII loading. AID is able to access the ssDNA generated during transcription by bacteriophage T7 polymerase *in vitro* but does so in a very strand-biased manner, predominantly targeting the looped-out non-template strand (Chaudhuri *et al*, 2003). However, during transcription *in vivo* this loop within the elongating transcription bubble is short, corresponding to an R-loop of 8 base pairs (Kireeva *et al*, 2000), and is likely

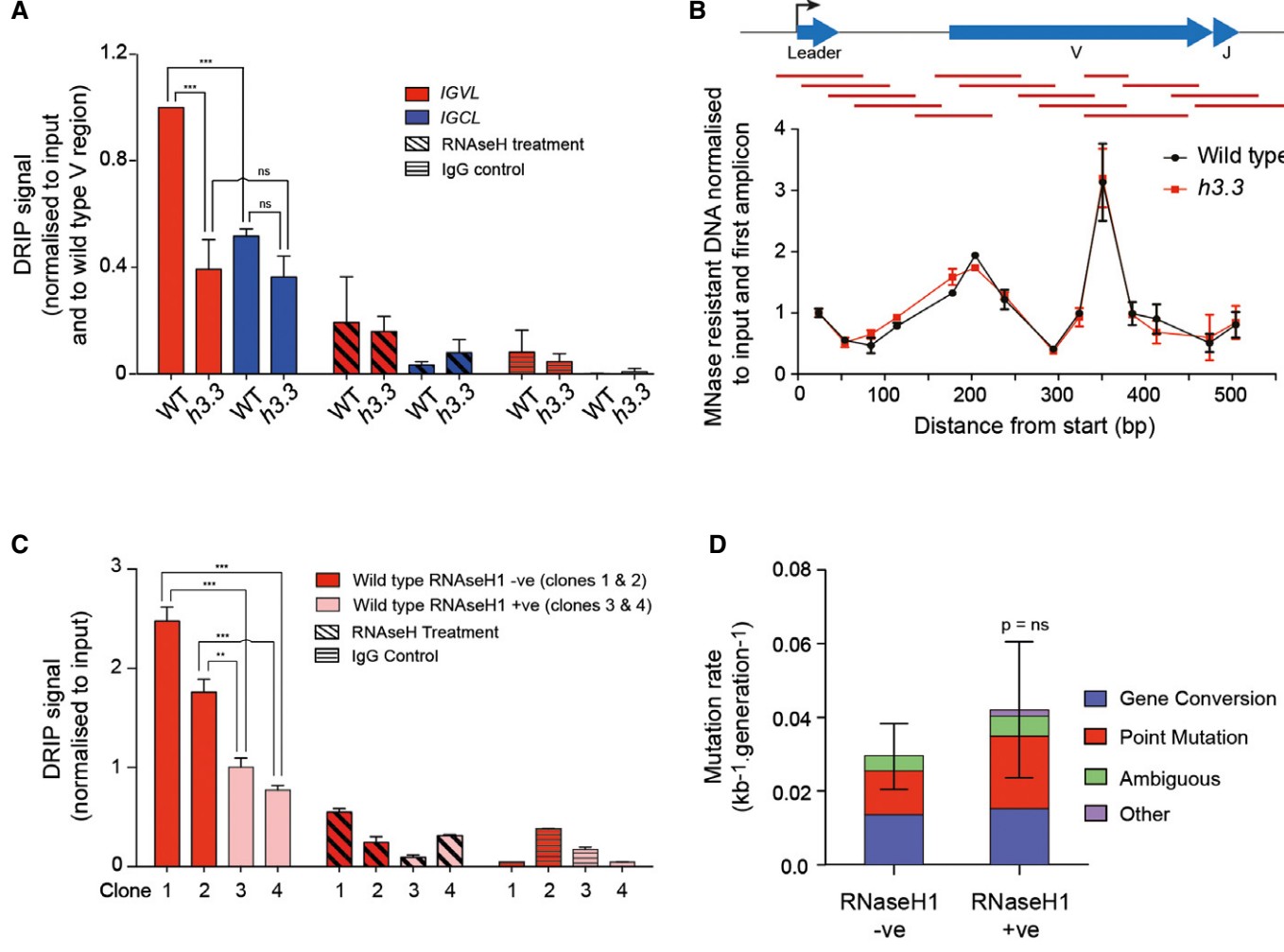

**Figure 4. H3.3 promotes formation of R-loops in *IGVL*.**

A DNA–RNA hybrids in *IGVL* and *IGCL* in wild type and *h3.3*. The DRIP signal is normalised to total input and to wild-type IgVλ. (***$P < 0.001$, ns = not significant, unpaired *t*-test for 3 biological replicates, error bars represent ± SD).

B MNase mapping reveals that loss of H3.3 does not affect average Igλ nucleosome positioning. The red bars indicate the qPCR amplicons for the MNase mapping. The qPCR signals are normalised to the first amplicon at the TSS. Error bars represent ± SD of 2 experimental repeats.

C DNA–RNA hybrids in *IGVL* of wild-type clones either without overexpression of RNase HI (red, clones 1 and 2, each assayed with three qPCR repeats) or with overexpression of RNase HI (pink, clones 3 and 4, each assayed with three qPCR repeats) (***$P < 0.001$, **$P < 0.01$; unpaired *t*-test). Expression of hAID^up^-FLAG and chicken RNase HI YFP in these clones is shown in Fig EV6. Error bars are ± SD of three biological replicates.

D Rate of Ig diversification in wild-type cells with and without overexpression of RNase HI. *P*-value calculated with a chi-square test. 96 sequences analysed for each condition. Error bars represent ± SD.

to be quite well protected within the RNAPII complex. It thus may not be readily accessible to AID under normal circumstances. However, pausing of RNAPII during transcription can lead to more extensive exposure of ssDNA in the transcription bubble (Gómez-González & Aguilera, 2007) and a number of studies have suggested that transcriptional pausing is also a key mechanism to attract AID to the IgV mutation domain (Kenter, 2012). Supporting this idea, AID has been physically linked to RNAPII through its ability to binding the RNAPII-associated processivity factor Spt5 (Pavri *et al*, 2010), the PAF complex (Willmann *et al*, 2012) and components of the exosome (Basu *et al*, 2011), the latter observation providing a potentially elegant solution to the problem of AID targeting both strands of DNA,

as the exosome could facilitate exposure of the template strand in addition to the coding strand.

An alternative, but not mutually exclusive, model for generating AID-accessible ssDNA is based on the formation of denaturation bubbles generated by transcription-associated negative supercoiling (Shen & Storb, 2004; Shen *et al*, 2005; Parsa *et al*, 2012). In regions of very active transcription, topoisomerase-mediated resolution of the negative supercoiling generated in the wake of RNAPII does not appear to be able to keep pace resulting in overall negative supercoiling upstream of RNAPII (Kouzine *et al*, 2008). Negative supercoiling creates localised denaturation bubbles (Jeon *et al*, 2010), which have been shown to provide an ideal substrate for AID on both strands of a plasmid, even in the

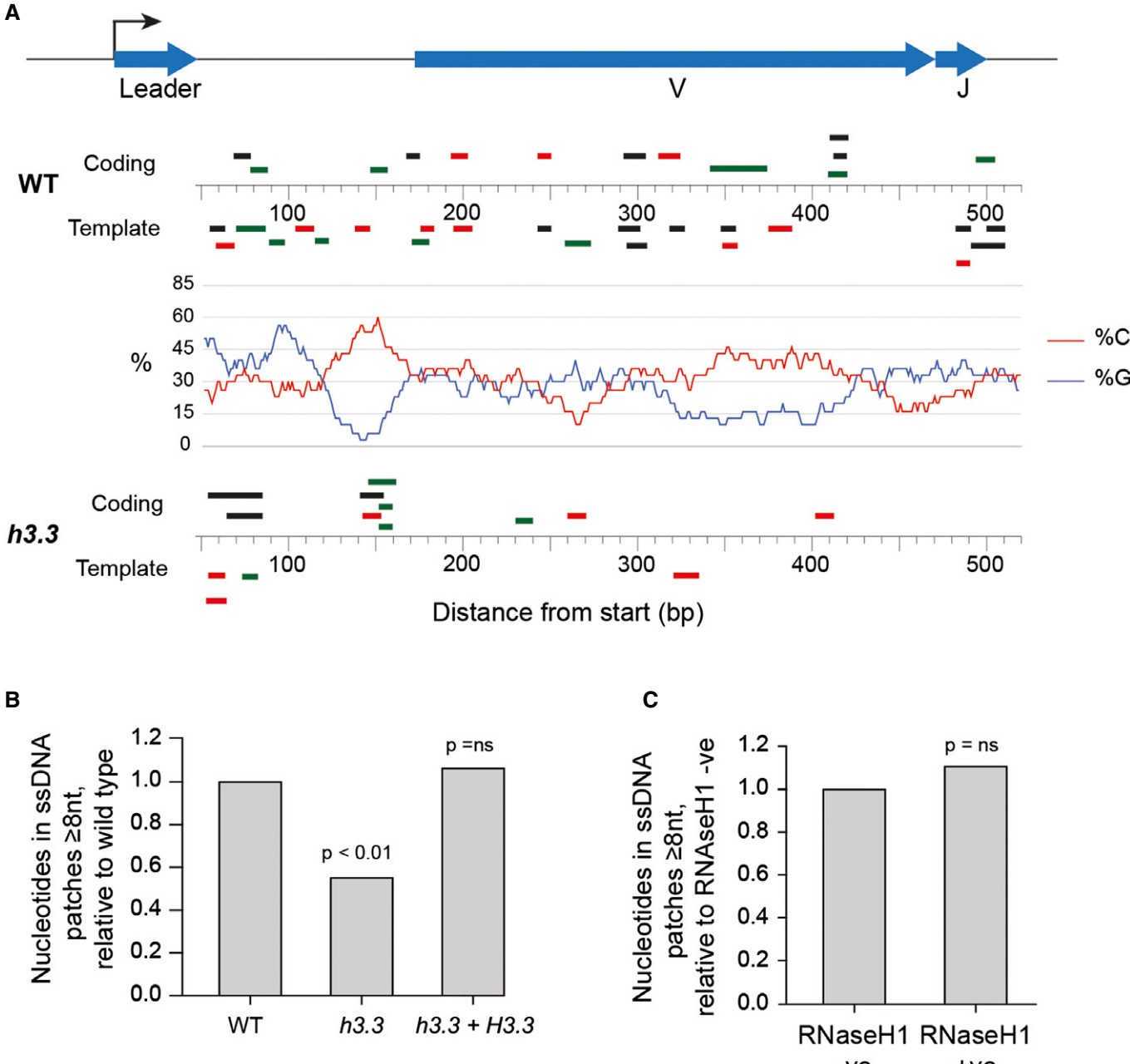

**Figure 5.  H3.3 promotes the formation of ssDNA stretches along the IgV locus.**

A    Location of ssDNA tracts ≥ 8 nt in *IGVL* in wild-type (WT) and *h3.3* cells detected by bisulphite sequencing. Data are from three independent experiments, the results from each being shown as black, red and green bars, respectively. ssDNA on the coding strand is shown above the marker line (numbered relative to the TSS at 0) and on the template strand below. In the centre, the percentage of G and C bases on the coding strand in a 30-bp rolling average window is shown.

B    Quantitation of the bisulphite detection of single-stranded DNA in wild-type, *h3.3* and *h3.3* complemented cells, normalised to wild type. *P*-values calculated with a chi-square test. Sequences analysed: wild type $n$ = 322, *h3.3* $n$ = 320 and *h3.3* + H3.3 $n$ = 95.

C    Bisulphite detection of single-stranded DNA in wild-type cells with and without RNase HI overexpression. *P*-value calculated with a chi-square test. Sequences analysed: RNase HI+ve $n$ = 94, RNase HI−ve $n$ = 90.

absence of transcription (Shen & Storb, 2004; Shen *et al*, 2005). Transcription-associated negative supercoiling has also been proposed to explain the short patches of AID and bisulphite-accessible ssDNA observed in the Ig genes of hypermutating Ramos cells and stimulated murine primary B cells (Ronai *et al*,

2007; Parsa *et al*, 2012), which are similar to the patches of ssDNA we observe in the diversifying *IGVL~R~* of DT40 and, importantly, show are sensitive to the absence of H3.3.

Exactly how these mechanisms relate to the ability of H3.3 to enhance the exposure of AID-accessible ssDNA remains unclear.

Nucleosome cores containing H3.3 are thought to associate less strongly with DNA than those containing canonical H3 variants (Jin & Felsenfeld, 2007; Thakar *et al*, 2009). Further, strong nucleosome positioning sequences lead to reduced Ig diversification compared with DNA sequences that only weakly support nucleosome formation (Kodgire *et al*, 2012). Synthesising these observations leads us to propose that the presence of H3.3 makes DNA of the Ig V gene more susceptible to ssDNA formation in the wake of transcription by altering the kinetic equilibrium between nucleosome loss and replacement. This would mean that while the overall average position of the nucleosomes is not altered, their displacement is favoured allowing more ready formation of ssDNA, which may also promote R-loop formation (Fig 6). It is thus plausible that the presence of H3.3 in the nucleosomes of the IgV genes could enhance the formation of ssDNA generated both by RNAPII pausing and the formation of supercoiling-induced denaturation bubbles (Fig 6).

Since H3.3 is enriched in specific genomic locations, including transcribed genes (Mito *et al*, 2005; Goldberg *et al*, 2010), the stimulation of AID access to DNA by H3.3 could help restrict AID activity to a relatively limited fraction of the genome and may thus contribute to preventing widespread off-target mutagenesis by AID. Nonetheless, H3.3 is clearly not sufficient to target AID since the majority of transcribed genes, which are enriched in H3.3, are not significant AID targets. Thus, the question still remains what is special about the Ig loci that promotes high levels of AID-mediated mutagenesis? The striking influence of nucleosome composition on Ig diversification does suggest to us the interesting possibility that local physical constraints on the locus could collaborate with H3.3-containing nucleosomes to promote ssDNA formation during transcription by enhancing the probability that negative supercoiling induces ssDNA bubbles. Such a model could potentially help explain the role of the matrix attachment regions in promoting strong Ig diversification (Betz *et al*, 1994). It will therefore be interesting in future studies

to determine the extent to which genomic structural features allow H3.3 to influence ssDNA exposure and mutagenesis genome-wide.

## Materials and Methods

### DT40 cell culture and mutants

DT40 cells were cultured at 37°C and 10% $CO_2$ in RPMI 1640 supplemented with 7% foetal calf serum as previously described (Simpson & Sale, 2003). The generation of H3.3-deficient (*h3.3*) DT40 cells was also reported previously (Frey *et al*, 2014). Complementation of *h3.3* cells was achieved by transfection of a vector driving expression of untagged chicken H3.3, followed by selection of puromycin-resistant clones. Expression of H3.3 was confirmed by qPCR.

### Chromatin immunoprecipitation

Chromatin immunoprecipitation (ChIP) was performed as previously described (Schiavone *et al*, 2014). Sonicated chromatin samples were incubated overnight at 4°C with the following antibodies: total histone H3 (Abcam, ab1791), histone H3.3 (Cell Signaling, 09-838), H3K4me3 (Cell Signaling, 9727), H3K36me3 (Abcam, ab9050). Normal rabbit IgG (Millipore) was used as a negative control. PCR primers for ChIP qPCR are listed in Table EV2.

### qPCR and RT–qPCR

All qPCR was performed using PowerSYBR® Green PCR Master Mix (Applied Biosystems) following the manufacturer's recommendations. RNA was extracted using the RNeasy® Mini Kit (Qiagen), and the preparation of cDNA was done using Quantitect® Reverse Transcription Kit (Qiagen) following the manufacturer's instructions. Where gene expression levels were determined, cDNA levels of genes of interest were normalised to the expression levels of EF1α within each sample.

### Analysis of IgVλ diversification

Wild-type and *h3.3* cells were transfected as previously described (Sale, 2006) with a pEAK8-based vector driving expression of an N-terminally FLAG-tagged, hyperactive variant of human AID, hAID[up] 7.3 (Wang *et al*, 2009). Transfectants were selected in medium containing neomycin, and individual clones picked and maintained in 24-well plate for 36 cell divisions. hAID[up] expression levels were determined during the last week of culture by qPCR and confirmed by Western blotting. Genomic DNA was extracted and amplified by PCR using Q5 Hot Start High-Fidelity DNA polymerase (NEB, M0493) and the following thermal protocol: 1 cycle at 98°C (1 min); 30 cycles at 98°C (15 s), 72°C (65 s); 1 cycle at 72°C (2 min). The rearranged Vλ locus was amplified with primers JSCVLF5 and JSCVLR3 (Table EV2) and cloned into pBluescript to generate a library as previously described (Sale *et al*, 2001). Sequence alignment and mutation scoring were performed as previously described (Sale *et al*, 2001).

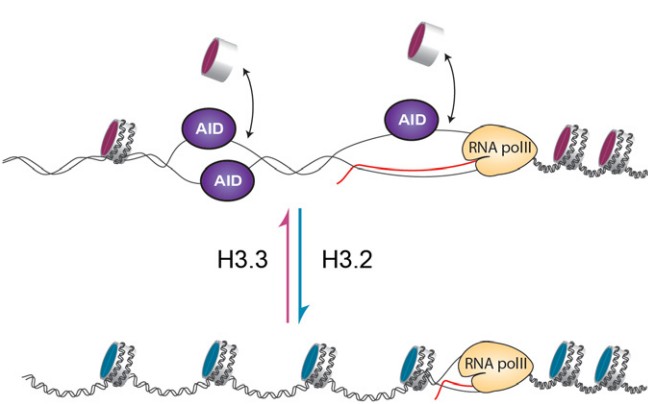

**Figure 6. Model summarising the proposed mechanism by which H3.3 stimulates AID-mediated Ig diversification.**

We suggest that H3.3-containing nucleosome cores (pink) are more likely to be dynamically exchanged than canonical H3-containing cores (blue) resulting in a greater probability that DNA can become single-stranded either as a result of pausing of RNAPII or of transcription-induced negative supercoiling generating transient denaturation bubbles.

## Fluctuation analysis for loss of surface IgM (sIgM) expression

Wild-type and *h3.3* cells were washed twice with PBS and stained for 20 min on ice with PE-conjugated mouse anti-chicken IgM antibody (Southern Biotech, clone M-1). After incubation, cells were washed with PBS and sIgM-positive cells sorted as single cells in 96-well plates with a MoFlo High Speed Cell sorter (Beckman Coulter). Individual clones were expanded for 36 divisions after cloning. Cells were stained with the IgM antibody and counterstained with SYTOX Blue (ThermoFisher Scientific) to monitor cell viability. The frequency of sIgM-loss variants monitored by flow cytometry using a gate fourfold below the mode of the sIgM-positive population.

## Inhibition of DT40 endogenous UDG activity

Wild-type and *h3.3* cells overexpressing hAIDup were transfected with the UGI-expression vector pEF-UGI (Di Noia & Neuberger, 2002). Transfectants were selected in medium containing puromycin, and individual clones picked and maintained in 24-well plates for 36 cell divisions. UGI-expression levels were determined the last week of culture by qPCR. Genomic DNA was extracted and used for library preparation and analysis of IgV gene mutations as described above.

## Western blotting

The following antibodies were used for immunoblotting: Flag-HRP (Sigma, A8592), H2A (Abcam, ab13923), GFP (Abcam, ab290), PCNA (Santa Cruz, sc-56), phospho-Chk1 (ser345) (Cell Signaling, 2348), γ-H2AX (Millipore, JBW301), β-actin (Thermo Scientific, AM4302), GAPDH (Sigma, G9545).

## Subcellular fractionation

Subcellular fractionation was carried out using the Subcellular Protein Fractionation Kit for cultured cells (Thermo Scientific, Ref. 78840). This kit generates a "soluble cytoplasmic fraction" by selective cell membrane permeabilisation. A total nuclear extract is generated from the intact nuclei by lysis of the nuclear membrane. The insoluble chromatin and chromatin-bound proteins are separated from the "soluble nuclear fraction" by centrifugation. The chromatin-bound proteins are released from the resulting pellet by treatment with micrococcal nuclease to generate the "chromatin fraction". The following modifications were made to the manufacturer's protocol: after isolation of the cytoplasmic fraction, the sample was clarified by centrifugation at 16,000 *g* for 15 min, and prior to extraction of the chromatin-bound fraction, the pellet was washed twice with NEB buffer.

## Nucleosome positioning assay

DT40 cells were washed with ice-cold PBS and incubated for 5 min in ice-cold NP-40 cell lysis buffer (10 mM Tris–Cl pH 7.4, 10 mM NaCl, 3 mM MgCl$_2$, 0.5% Nonidet P-40, 0.15 mM spermine, 0.5 mM spermidine). Cells were washed in MNase digestion buffer (10 mM Tris–Cl pH 7.4, 15 mM NaCl, 60 mM KCl, 0.15 mM spermine and 0.5 mM spermidine) and then

resuspended in MNase digestion buffer supplemented with 1 mM CaCl$_2$ to a final density of $100 \times 10^6$ cells/ml. Two hundred microlitres of the cellular dilution was incubated with or without (total input) 80 U of MNase (Thermo Scientific) for 30 min at 37°C, after which the reaction was stopped by adding 160 μl of MNase digestion buffer and 66 μl of stop buffer (60.6 mM EDTA, 6 mM EGTA, 6% SDS and 2.27 mg/ml of proteinase K). The samples were phenol–chloroform-extracted, treated with 20 μg of RNase for 2 h at 37°C, phenol–chloroform-extracted again and washed by ethanol–acetate precipitation. The MNase-resistant DNA was quantified by qPCR using the primer pairs detailed in Table EV2.

## Bisulphite mapping of single-stranded DNA

Bisulphite mapping of single-stranded DNA was performed as previously described (Ronai *et al*, 2007) with the following modifications: incubations in buffers A, B and C were carried out for 10 min; nuclei were resuspended in Promega Nuclear Lysis Solution before being incubated in 2.5 M sodium bisulphite and 20 mM hydroquinone. After DNA purification, the IgVλ region was amplified with ZymoTaq™ DNA polymerase (Zymo Research), using primers Bisulfite_Map_Fw and Bisulfite_Map_Rev (Table EV2). The resulting fragments were cloned into pBluescript vector and sequenced. The length of the single-stranded DNA patches was determined by counting the distance, in nucleotides, between the two most distant consecutive dCs that have been converted to dTs (for ssDNA in the coding strand, read as dGs to dAs for the template strand), including these converted nucleotides and all the non-dCs (dGs for the template strand) enclosed within the segment. Since the normal transcription bubble is thought to be 8 nt in length (Kireeva *et al*, 2000), only segments of a length of ≥ 8 nucleotides were included in the analysis. The plot of dG and dC content along *IGVL*$_R$ was created using the GC Content Calculator tool (http://www.biologicscorp.com/tools/GCContent/), with a window size of 30 nucleotides.

## *In vitro* AID activity assay

Crude protein extracts were prepared from $8 \times 10^7$ cells by a 30-min incubation on ice in lysis buffer (50 mM Tris–HCl pH 7.4, 150 mM NaCl, 20 mM EDTA, 50 mM NaF, 1% (v/v) Triton X-100 plus protease and phosphatase inhibitors (Roche)). hAID$^{up}$ was pulled down with anti-FLAG M2 affinity gel (Sigma-Aldrich, A2220) and eluted by competition in non-denaturing conditions with Flag peptide (Sigma-Aldrich, F4799). Deaminase activity of purified hAID$^{up}$ was assayed following the protocol reported in Wang *et al* (2009) with the following modifications: 600 ng of purified protein were incubated at 37°C for different times in 20 μl of reaction buffer (20 mM Tris–HCl pH 8, 1 mM EDTA, 1 mM DTT, 0.5 U UDG (NEB), 10 μg RNase A) and 1 pmol of fluoresceine-labelled AID_Activity oligo (Table EV2). Samples were incubated for 30 min at 95°C in the presence of 30 mM NaOH and then neutralised by the addition of 30 mM HCl. Samples were subsequently mixed with an equal volume of loading dye (95% formamide, 1 mM EDTA) and resolved on 10% TBE-urea gels (Life Technologies). Fluorescence was detected with a Typhoon PhosphorImager (GE Life Science) and the extent of

deamination determined as previously described (Wang *et al*, 2009).

## DNA/RNA immunoprecipitation (DRIP)

DNA/RNA hybrids were immunoprecipitated with a monoclonal antibody purified from the S9.6 hybridoma cell line following the protocol described in the study of Groh *et al* (2014) with a few modifications. Briefly, after extraction and lysis of nuclei, genomic DNA containing R-loops was digested overnight with the following mixture of restriction enzymes: BamHI, NcoI, PvuII, ApaLI and NheI. Digested genomic DNA was then diluted in 5 ml IP dilution buffer and pre-cleared with 30 μl of Protein-G Sepharose beads (Roche) for 1 h at 4°C. Pre-cleared genomic DNA was immunoprecipitated overnight at 4°C with 10 μg of S9.6 antibody. Subsequent washes and elution steps are identical to the procedure as described for ChIP. RNase H sensitivity assay was performed by incubating digested genomic DNA with 25 U RNase H (NEB, M0297S) as previously described (Groh *et al*, 2014). Immunoprecipitated DNA/RNA hybrids were quantified by qPCR using the primers listed in Table EV2. The signal was normalised to total input and reported relative to wild-type cells. IgG-only immunoprecipitations were performed as background signal control.

## Measurement of transcription elongation speed of RNA polymerase II by 4sUDRB

RNA Pol II elongation rate was measured taking advantage of the reversible inhibition of transcription elongation by 5,6-dichloro-1-β-D-ribofuranosylbenzimidazole (DRB) combined with a pulse of 4-thiouridine (4sU) as described (Fuchs *et al*, 2015) with some modifications to increase the incorporation efficiency for ultra short labelling. Briefly, $15 \times 10^7$ cells were immobilised for 1 h on 150-mm tissue culture dishes pre-coated with 0.01% (w/v) poly-L-lysine (Sigma-Aldrich, P8920) and then treated with 100 μM DRB for 3 h. Eight minutes before completion of the 3-h DRB treatment, 1.5 mM 4sU was added directly to the dishes. After completion of the 3-h incubation, cells were washed quickly with pre-warmed PBS and incubated for variable times with complete medium containing 1.5 mM 4sU. At the end of the pulse labelling period, cells were lysed and 200 μg of total RNA was biotinylated and enriched on streptavidin-coupled beads as previously described (Fuchs *et al*, 2015). Enrichment of cDNA generated from 4sU-tagged RNA was evaluated by qPCR using the primers reported in Table EV2.

## Data availability

Referenced data: Frey A, Listovsky T, Guilbaud G, Sarkies P, Sale JE (2014). RNA-sequencing for *h3.3* cells. ArrayExpress accession number EMTAB-2754.

**Expanded View** for this article is available online.

## Acknowledgements

We thank Sasa Svikovic for the YFP-RNaseHI plasmid and advice on DRIP, Matthew Scharff for advice on bisulphite mapping, Maria Daly and Fan Zhang in the LMB flow cytometry facility and Cristina Rada and Gareth Williams for comments and advice. This study was funded by a Medical Research Council grant to LMB (U105178808), by a César Milstein Studentship from the Darwin Trust of Edinburgh (to MR) and a Jürgen Manchot Stiftung studentship (to AF).

## Author contributions

JES, MR and DS designed the study and wrote the manuscript, and MR, DS and AF carried out the experiments.

## Conflict of interest

The authors declare that they have no conflict of interest.

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
