## [Review Process File · The EMBO Journal]

Manuscript EMBO-2016-93958

Histone H3.3 promotes IgV gene diversification by enhancing formation of AID-accessible single stranded DNA

Marina Romanello, Davide Schiavone, Alexander Frey and Julian E. Sale

Corresponding author: Julian E. Sale, MRC Laboratory of Molecular Biology

Review timeline:

Submission date:	25 January 2016
Editorial Decision:	17 February 2016
Revision received:	07 April 2016
Accepted:	25 April 2016

*Editor: Hartmut Vodermaier***Transaction Report:**

1st Editorial Decision

17 February 2016

Thank you again for submitting your manuscript on H3.3 roles in AID-mediated IgV gene diversification to The EMBO Journal. We have now received reviews from three expert referees, copied below for your information. Overall, all referees appreciate the interest and potential importance of your results, but they also point out several important caveats, as you will see from their reports. During our pre-decision cross-commenting session, the reviewers further discussed the key issues with each other, with the conclusions of these discussions summarized here:

- there are concerns regarding the relative importance of H3.3 as compared to transcription elongation/stalling, which may have led to some initial misunderstanding on the part of referee 1 but which clearly may require some better discussion (see referees 1 and 3)
- the R-loop formation data remain inconclusive in the absence of RNase A treatment in the DRIP assays (see referee 2)
- major concerns relate to the use of hyperactive AID mutants and the unclear relevance of the derived findings in the physiological (wt AID) context (see referees 1 and 2). The referees appreciate the use of mutant AID as an investigational tool to uncover H3.3 functions in chromatin, but it would have to be clearly reflected in the writing if this was the main focus of the paper. The scope and direct relevance of the paper for the AID field would however be greatly increased by repeating at least some of the basic experiments also with wt AID, and adjusting of title and conclusions depending on the outcome and phenotypes.

Based on these comments, I would like to give you an option to revise the manuscript, but I need to point out that the decision regarding eventual acceptance for The EMBO Journal will depend on successful validation of the R-loop assays and extension to wt AID contexts. In this light, I would encourage you here to send us a brief proposal (in the form of a tentative response letter) on how you might be able to answer the referees' comments; this would allow us to clarify the feasibility of the proposed revision work, and to define which improvements would be key for eventual

acceptance in The EMBO Journal. We could further arrange for an extended revision period, during which time the publication of any competing work elsewhere would as usual have no negative impact on our final assessment of your own study.

Thank you again for the opportunity to consider this work for publication, and I look forward to hearing from you.

REFEREE COMMENTS

Referee #1:

Antigen driven antibody diversification mechanisms of class switch recombination, somatic hypermutation and gene conversion are absolutely dependent upon the DNA deamination activity of the protein Activation Induced Deaminase (AID). The transcription-associated mechanisms that AID utilizes to hypermutate DNA in the Immunoglobulin loci of B cells has been now being worked out. AID hypermutates single-strand DNA structures that are cotranscriptionally generated by physically associating with the transcribing RNA polymerase II complex. Thus, many mechanisms such as RNA polII pausing, splicing, RNA degradation that are intimately coupled with transcription are also important for AID's activity in vivo. In this manuscript, the authors have performed experiments to claim that ssDNA structures that are targets of AID are not associated with RNA polymerase II and its transcriptional state, but rather due to the presence of a histone variant H3.3 at IgV DNA sequences in the chicken DT40 cell line. This could be important but the basis of delineating H3.3 dependent ssDNA formation from the RNA polII elongation complex (and the associated stalling/pausing mechanisms) are weak and unwarranted. Moreover the conclusions drawn from using the AID mutants in this manuscript should be drawn with caution. The authors may consider re-addressing these points experimentally or rewriting the manuscript to align their conclusions with the data they provide (and other laboratories have extensively provided). Some important points to consider are:

1. The use of the AID up-mutant to establish AID activity is quite artificial. These are proteins that are either altered for substrate specificity or for AID's catalytic site. Without understanding the mechanism of the AIDup mutant activity, using it for AID's mechanistic analysis is risky. Moreover, the expression of the WT AID protein demonstrates no difference (of statistical significance) in percentage of sIgM loss between WT and H3.3 mutant cells (Fig. 1A). Some explanation is required.
2. The rest of the Fig. 1 represents data obtained with AIDup mutant and should be labelled accordingly.
3. Experiments in Figure 2 were performed with AIDup mutant. The AID dependent mutation frequency in DT40 cells have been previously analyzed, it is not clear why the authors need to use AIDup mutant for these experiments.
3. The authors should be cautious about the interpretation of Fig. 3E. The model relating RNA polII stalling/pausing with AID activity proposes that only a very small fraction of transcribing RNA polII stalls and associates with AID to cause somatic DNA mutagenesis. The conditions that cause this stalling event is not known and can be a combination of many aspects including histone modifications, secondary DNA structures, non-coding RNA expression etc. However, only a very minor fraction of RNA polII needs to stall, to catalyze recruitment of AID on substrate DNA. In Fig. 3E, the authors assume that a wholesome change in RNA polII elongation rates would occur in cells that are defective in somatic mutagenesis. This is incorrect and thus authors either need to perform more experiments to make the conclusion or rephrase statements like "Thus, H3.3 does not influence RNAPII elongation suggesting that altered pausing of the polymerase is unlikely to explain the observed differences in the recruitment of AID to IGVL seen in h3.3 cells."

Referee #2:

This manuscript describes the analysis of Ig V region diversification in DT40 cells lacking histone H3.3. The authors find that, in cells overexpressing a hyperactive AID mutant, the absence of H3.3 results in a substantial drop in IgV region mutation. The rest of the manuscript focuses on

understanding this phenotype. The authors show that the defect affects both SHM and GCV, operates at the level of AID action (and not a later repair step), and is not due to a discernible change in transcription of the locus. Nor does altered nucleosome positioning appear to explain the phenotype. The authors do find that single stranded DNA bubbles are decreased in IgV in the absence of H3.3, and they do a nice job of arguing against an essential role for stable R loops in the h3.3 phenotype using RNaseH-expressing cells. The Discussion is interesting, arguing for a model in which H3.3 facilitates the formation of single stranded regions by increasing the propensity of the nucleosome to dissociate, perhaps under the stress caused by supercoiling. This model is interesting and links to previous findings in the field in plausible ways.

No histone isoform has previously been implicated in SHM/GCV of IgV regions previously, and the findings, if relevant to the normal action of AID, are interesting and important. The manuscript has one major problem and a few minor issues worth addressing.

The major problem arises at the very beginning, in Figure 1A: while a clear defect in GCV/SHM is seen with the overexpressed hyperactive mutant AID, a much weaker defect (if any at all) is seen with WT AID. The authors do not address this discrepancy in the text or experimentally, and it is a big problem because it leaves open the worry that the phenotype would not be seen with WT AID and hence is not relevant to the physiological process of AID-mediated V region diversification. I am open to the possibility that the hyperactive mutant AID reveals a phenotype that is harder to tease out with WT AID, but the authors need to do a lot more to address this. One way of looking at the data is that H3.3 makes only a minor contribution to the action of AID, with the h3.3 phenotype being much exaggerated by the hyperactive mutant. This concern does not extend to the notable finding of a 2-fold drop in single stranded bubbles in the absence of H3.3 (a key finding the authors should probably strengthen with data from the H3.3-reconstituted cells). My point is that one would really like to know just how much of a defect in GCV/SHM this translates into with WT AID. I'm not so much worried about AID overexpression per se, since WT DT40 expresses pretty low levels of AID (likely well below those seen in diversifying B cells in vivo), but I am worried about the use of the hyperactive mutant.

Minor issues

1. The reconstitution experiment with H3.3 (Figure 1B) is an important one. The authors should also show IgM loss data for the H3.3-reconstituted line, as in Figure 1A.
2. There does appear to be a reduction in AID in the chromatin bound fraction in h3.3 cells compared to WT (Fig. EV3). The authors conclude that AID subcellular localization is not "grossly altered", which I agree with; nonetheless, the small reduction seen could be a significant part of the phenotype of h3.3 cells and deserves comment. Also, it would be helpful for readers (certainly this reader) to explain how the subcellular fractionation performed here differs from the standard nuclear/cytoplasmic fractionation that is routinely done. Does this method only capture AID that is chromatin bound, excluding any AID that is freely diffusible in the nucleus? What is seen with a more typical fractionation? These are important points because small changes in AID nuclear localization could have a substantial effect on activity and could indicate that some additional mechanisms are at play.
3. The central plot in Figure 5A: which strand of the DNA is being analyzed for percent C and G?

Referee #3:

A key question regarding transcription is the role of histones, histone variants and covalent histone modifications on transcription mechanics. The function of histone H3 variant H3.3 which differs by only 4 aa from canonical H3 is largely unknown. It has been observed that nucleosomes containing H3.3 are unusually sensitive to salt dependent disruption suggesting that introduction of H3.3 into nucleosomes could lead to increased chromatin accessibility. Romanello and coworkers have examined the role of histone H3.3 in creating chromatin accessibility (and ssDNA) in chicken DT40

cells by abolishing H3.3 expression by targeted deletion as reported in previous work. In this manuscript, they have examined the role of H3.3 by leveraging the requirement of activation induced deaminase (AID) for ssDNA substrate and using the rate of mutation acquisition at the IgV locus via AID as a well characterized system of inquiry. This well written manuscript reports that H3.3 deletion impairs AID dependent mutagenesis of IgV templates without discernable impact on steady state transcription, elongation rates in DRB assays and histone modifications. AID localization and enzymatic activity is unaffected by H3.3 knockout confirming that reduced mutagenesis does not occur due to perturbed AID activity. They also show that nucleosome positioning remains unaltered in H3.3 KO cells. The controls for the integrity of transcription and AID functions were thorough and convincing.

The authors provide evidence that in H3.3 knockout cells R-loop formation is reduced as assessed using DNA-RNA hybrid IP (DRIP) assays and the S9.6 Ab (see Figure 4). They then show that reduction of R-loops *in vivo* has no effect on IgV mutagenesis. However, there are significant technical concerns with their DRIP assays (see below for details), which requires attention before definitive conclusions can be made. This is an important point since R-loop formation has been shown to modulate the efficiency of class switch recombination. Analysis of ssDNA using bisulfite sequencing indicates that H3.3 deficiency leads to reduced short patch ssDNA formation and provides a plausible mechanism to explain reduced AID dependent mutagenesis in h3.3 DT40 cells. However, the authors should be more circumspect about the discussion of their findings as the conclusions are based on a single H3.3 KO chicken cell line.

Specific Comments

Pg 7 line 12: The authors note that h3.3 deficiency led to little change in gene expression. Please add the reference for this.

Pg 9 line 19 "...activity of AID unaffected.." Please add in "is unaffected"

Pg9 line 22 A reference for R-loops is required.

Pg 9 line 18 and Pg 21 Methods. The authors use the DNA-RNA hybrid IP (DRIP) assay with the S9.6 antibody to assess the extent of ssDNA formation in the presence and absence of H3.3. However, recent work has shown that S9.6 Ab binds AU-rich RNA:RNA duplexes with a KD that is only 5.6-fold weaker than for RNA:DNA duplexes (Zhang et al BMC Res Notes. 2015 Apr 8;8:127. doi: 10.1186/s13104-015-1092-1). It has been reported that fold back of ssRNA can generate RNA:RNA duplexes that may bind the S9.6 antibody, and adventitious binding of RNA may produce short RNA:DNA regions (Zhang et al). Thus, use of the S9.6 antibody should be preceded by RNase A treatment to remove free ssRNA. The authors must be cautious when interpreting S9.6 data, and confirmation by independent structural and functional methods is essential.

Pg 10 line

Results using DNA-RNA hybrid IP (DRIP) with the S9.6 antibody are provided. However, there was no discussion in the text of the results regarding Figure 4A for samples that were treated with RNaseH.

Pg 10 line 16 "therefore" should be deleted.

Pg 11 line 4: The authors conclude that short patches of ssDNA are unaffected by RNaseH treatment and therefore are not related to the presence of R-loops since RNaseH treatment abolished R-loops but not ssDNA patches. It is not clear how one would know whether R-loops are a consequence of ssDNA by means of the RNaseH experiment they provide. It is possible that RNaseH sensitive R-loops originate from RNaseH refractory small ssDNA patches.

Pg 13-14: The discussion of pausing as an alternate mechanism AID mutagenesis is somewhat misleading. RNAPII pausing is most likely a mechanism for AID recruitment through Spt5 and most likely contributes to R-loop formation and creation of ssDNA as substrate for AID. These issues are

mechanistically linked. The R-loop formation may be short patch or long stretches (as in S regions). The authors' discussion of these issues needs clarity.

Pg 15: The authors write, "The promotion of AID access by H3.3 would restrict AID activity to a relatively limited fraction of the genome and may thus contribute to preventing off-target mutagenesis. However, since H3.3 is enriched in many transcribed genes, most of which are not targeted by AID, the question still remains what is special about the Ig loci that promotes high levels of mutagenesis?" It is entirely possible that AID off-targeted genes are also enriched for H3.3 which in turn mediates formation of ssDNA patches. Do the authors know that AID off targeted genes lack H3.3? This section should be re-written.

1st Revision - authors' response

07 April 2016

Romanello et al. Response to the referee's comments

Referee #1:

Antigen driven antibody diversification mechanisms of class switch recombination, somatic hypermutation and gene conversion are absolutely dependent upon the DNA deamination activity of the protein Activation Induced Deaminase (AID). The transcription-associated mechanisms that AID utilizes to hypermutate DNA in the Immunoglobulin loci of B cells has been now being worked out. AID hypermutates single-strand DNA structures that are cotranscriptionally generated by physically associating with the transcribing RNA polymerase II complex. Thus, many mechanisms such as RNA polII pausing, splicing, RNA degradation that are intimately coupled with transcription are also important for AID's activity in vivo. In this manuscript, the authors have performed experiments to claim that ssDNA structures that are targets of AID are not associated with RNA polymerase II and its transcriptional state, but rather due to the presence of a histone variant H3.3 at IgV DNA sequences in the chicken DT40 cell line. This could be important but the basis of delineating H3.3 dependent ssDNA formation from the RNA polII elongation complex (and the associated stalling/pausing mechanisms) are weak and unwarranted. Moreover the conclusions drawn from using the AID mutants in this manuscript should be drawn with caution. The authors may consider re-addressing these points experimentally or rewriting the manuscript to align their conclusions with the data they provide (and other laboratories have extensively provided).

We do not claim that the ssDNA formation and AID targeting is *not* associated with RNAPII. AID activity *in vivo* clearly requires transcription. However, our data shows that the presence or absence of H3.3 does not lead to a detectable *change* in *IGVL* transcription or RNAPII dynamics. Importantly, our data does not exclude stalling or pausing of RNAPII as a source of AID-accessible ssDNA. We have revised our discussion of these points to ensure this is clear.

Some important points to consider are:

1. The use of the AID up-mutant to establish AID activity is quite artificial. These are proteins that are either altered for substrate specificity or for AID's catalytic site. Without understanding the mechanism of the AIDup mutant activity, using it for AID's mechanistic analysis is risky.

We made a mistake here by not stating that we had made the original observations in cells expressing AID naturally and by not showing this data more clearly. The data from cells with wild type AID expression were, in fact, present in Figure 1C, which shows hAID^{up} expression *relative to wild type* (plotted at '1' in the graph). However, we appreciate that this was not an adequate representation of this data. We now show the sequencing data from cells without any AID overexpression in Figure 1 of the revised manuscript, which clearly demonstrates a reduction in Ig diversification in two independent *h3.3* mutants.

We used the AID^{up} mutant to bolster this initial observation by demonstrating the difference between wild type and *h3.3* diversification at a range of deamination levels. Although more active, the spectrum of deamination by the AID^{up} mutant does not differ from the wild type enzyme (Wang et

al, 2009; Taylor et al, 2013). It thus provides a very useful tool in this context.

Moreover, the expression of the WT AID protein demonstrates no difference (of statistical significance) in percentage of sIgM loss between WT and H3.3 mutant cells (Fig.1A). Some explanation is required.

Actually, the reduction in the formation of sIg loss variants in *h3.3* cells compared with wild type, shown in the original Figure 1A, is statistically significant at $p < 0.01$ (Mann-Whitney test). We apologise for not including the statistic in the original figure. We have carefully reanalysed the data using more stringent gating and the difference remains. However, in my experience, this assay is not that sensitive in DT40 cells diversifying predominantly by gene conversion without AID overexpression. The rate of apparent sIg loss variant generation in *h3.3* is close to the background of the assay. The fluctuation analysis can be interpreted more clearly in the context of the AID^{up} overexpression, but the most reliable evidence comes from the sequencing data from cells with multiple levels deamination activity, including with natural AID expression (Figures 1A, 1B, 1E and 1F).

2. The rest of the Fig.1 represents data obtained with AIDup mutant and should be labelled accordingly.

Figure 1 was clearly labelled showing that the data referred to the AID^{up} mutant. We have now added data from cells with wild type AID expression, and clearly differentiate this from the AID^{up} mutant data.

3. Experiments in Figure 2 were performed with AIDup mutant. The AID dependent mutation frequency in DT40 cells have been previously analyzed, it is not clear why the authors need to use AIDup mutant for these experiments.

The use of the AID^{up} mutant as a means of testing whether the role of H3.3 is at the level of AID access to IgV or the processing of dU is appropriate and gives a clear result that allows us to differentiate a role for H3.3 before or after dU formation.

3. The authors should be cautious about the interpretation of Fig. 3E. The model relating RNA polII stalling/pausing with AID activity proposes that only a very small fraction of transcribing RNA polII stalls and associates with AID to cause somatic DNA mutagenesis. The conditions that cause this stalling event is not known and can be a combination of many aspects including histone modifications, secondary DNA structures, non-coding RNA expression etc. However, only a very minor fraction of RNA polII needs to stall, to catalyze recruitment of AID on substrate DNA. In Fig. 3E, the authors assume that a wholesome change in RNA polII elongation rates would occur in cells that are defective in somatic mutagenesis. This is incorrect and thus authors either need to perform more experiments to make the conclusion or rephrase statements like "Thus, H3.3 does not influence RNAPII elongation suggesting that altered pausing of the polymerase is unlikely to explain the observed differences in the recruitment of AID to IGVL seen in h3.3 cells."

We did not assume that we would observe a wholesale change in RNAPII elongation rates. However, the fact that transcriptional rates or levels are not affected by loss of H3.3 is an important observation as it shows that the reduction in Ig diversification is not simply due to reduced transcription. Importantly, our data do not exclude RNAPII stalling as a mechanism for AID recruitment but suggest that it takes place in both wild type and *h3.3* cells at the same frequency, or that any change is below the sensitivity threshold of our assays. We have clarified the discussion of this point and the relationship between the effect of H3.3 and existing models.

Referee #2:

This manuscript describes the analysis of Ig V region diversification in DT40 cells lacking histone H3.3. The authors find that, in cells overexpressing a hyperactive AID mutant, the absence of H3.3 results in a substantial drop in IgV region mutation. The rest of the manuscript focuses on understanding this phenotype. The authors show that the defect affects both SHM and GCV, operates at the level of AID action (and not a later repair step), and is not due to a discernible

change in transcription of the locus. Nor does altered nucleosome positioning appear to explain the phenotype. The authors do find that single stranded DNA bubbles are decreased in IgV in the absence of H3.3, and they do a nice job of arguing against an essential role for stable R loops in the h3.3 phenotype using RNaseH-expressing cells. The Discussion is interesting, arguing for a model in which H3.3 facilitates the formation of single stranded regions by increasing the propensity of the nucleosome to dissociate, perhaps under the stress caused by supercoiling. This model is interesting and links to previous findings in the field in plausible ways.

No histone isoform has previously been implicated in SHM/GCV of IgV regions previously, and the findings, if relevant to the normal action of AID, are interesting and important. The manuscript has one major problem and a few minor issues worth addressing.

The major problem arises at the very beginning, in Figure 1A: while a clear defect in GCV/SHM is seen with the overexpressed hyperactive mutant AID, a much weaker defect (if any at all) is seen with WT AID. The authors do not address this discrepancy in the text or experimentally, and it is a big problem because it leaves open the worry that the phenotype would not be seen with WT AID and hence is not relevant to the physiological process of AID-mediated V region diversification. I am open to the possibility that the hyperactive mutant AID reveals a phenotype that is harder to tease out with WT AID, but the authors need to do a lot more to address this. One way of looking at the data is that H3.3 makes only a minor contribution to the action of AID, with the h3.3 phenotype being much exaggerated by the hyperactive mutant.

As discussed above, we did make the original observation using cells expressing AID naturally and observed a very striking decrease in Ig diversification. We did not make this at all clear in the original manuscript and have now included this data in Figure 1. However, we felt that basing all the work on natural AID expression would be unconvincing. Hence we chose to use overexpression of the AID^{up} mutant as a tool to explore the phenomenon more robustly and over a range of deamination levels.

This concern does not extend to the notable finding of a 2-fold drop in single stranded bubbles in the absence of H3.3 (a key finding the authors should probably strengthen with data from the H3.3-reconstituted cells).

We have now done this, and H3.3 reconstitution restores the frequency of ssDNA patches to wild type levels. These data are now included in Figure 5.

My point is that one would really like to know just how much of a defect in GCV/SHM this translates into with WT AID. I'm not so much worried about AID overexpression per se, since WT DT40 expresses pretty low levels of AID (likely well below those seen in diversifying B cells in vivo), but I am worried about the use of the hyperactive mutant.

I hope that the data we have now added to Figure 1 allays this concern. The reduction in Ig diversification in when H3.3 is absent is clearly seen in cells with endogenous AID expression.

Minor issues

1. The reconstitution experiment with H3.3 (Figure 1B) is an important one. The authors should also show IgM loss data for the H3.3-reconstituted line, as in Figure 1A.

We have now added this data to Figure 1, the result being consistent with the sequencing data in showing a restoration of diversification.

2. There does appear to be a reduction in AID in the chromatin bound fraction in h3.3 cells compared to WT (Fig. EV3). The authors conclude that AID subcellular localization is not "grossly altered", which I agree with; nonetheless, the small reduction seen could be a significant part of the phenotype of h3.3 cells and deserves comment. Also, it would be helpful for readers (certainly this reader) to explain how the subcellular fractionation performed here differs from the standard nuclear/cytoplasmic fractionation that is routinely done. Does this method only capture AID that is chromatin bound, excluding any AID that is freely diffusible in the nucleus? What is seen with a

more typical fractionation? These are important points because small changes in AID nuclear localization could have a substantial effect on activity and could indicate that some additional mechanisms are at play.

The 'chromatin' fraction distinguishes the protein that is associated with DNA rather than that which is free in the nucleus. It relies on micrococcal nuclease digestion to release proteins bound to chromatin. We have now explained this in detail in the methods. We are reluctant to make much too much of this as this is a crude assay with overexpressed AID protein. We have repeated the experiment and included a quantification of the results in Figure EV3. We do not see any clear difference in the distribution of AID in wild type and *h3.3*. We thus think that it is very unlikely that a gross change in AID localisation is the explanation for the differences in Ig diversification. In any event, this would not explain the significantly diminished ssDNA exposure in the absence of H3.3.

3. The central plot in Figure 5A: which strand of the DNA is being analyzed for percent C and G?

The coding strand. We have now made this clear in the figure legend.

Referee #3:

A key question regarding transcription is the role of histones, histone variants and covalent histone modifications on transcription mechanics. The function of histone H3 variant H3.3 which differs by only 4 aa from canonical H3 is largely unknown. It has been observed that nucleosomes containing H3.3 are unusually sensitive to salt dependent disruption suggesting that introduction of H3.3 into nucleosomes could lead to increased chromatin accessibility. Romanello and coworkers have examined the role of histone H3.3 in creating chromatin accessibility (and ssDNA) in chicken DT40 cells by abolishing H3.3 expression by targeted deletion as reported in previous work. In this manuscript, they have examined the role of H3.3 by leveraging the requirement of activation induced deaminase (AID) for ssDNA substrate and using the rate of mutation acquisition at the IgV locus via AID as a well characterized system of inquiry. This well written manuscript reports that H3.3 deletion impairs

AID dependent mutagenesis of IgV templates without discernable impact on steady state transcription, elongation rates in DRB assays and histone modifications. AID localization and enzymatic activity is unaffected by H3.3 knockout confirming that reduced mutagenesis does not occur due to perturbed AID activity. They also show that nucleosome positioning remains unaltered in H3.3 KO cells. The controls for the integrity of transcription and AID functions were thorough and convincing.

*The authors provide evidence that in H3.3 knockout cells R-loop formation is reduced as assessed using DNA-RNA hybrid IP (DRIP) assays and the S9.6 Ab (see Figure 4). They then show that reduction of R-loops in vivo has no affect on IgV mutagenesis. However, there are significant technical concerns with their DRIP assays (see below for details), which requires attention before definitive conclusions can be made. This is an important point since R-loop formation has been shown to modulate the efficiency of class switch recombination. Analysis of ssDNA using bisulfite sequencing indicates that H3.3 deficiency leads to reduced short patch ssDNA formation and provides a plausible mechanism to explain reduced AID dependent mutagenesis in *h3.3* DT40 cells. However, the authors should be more circumspect about the discussion of their findings as the conclusions are based on a single H3.3 KO chicken cell line.*

We have added Ig sequencing data from a second, independent *h3.3* line to Figure 1.

Specific Comments

*Pg 7 line 12: The authors note that *h3.3* deficiency led to little change in gene expression. Please add the reference for this.*

Done. This is shown in our previous paper on the H3.3 knockout DT40 cells (Frey et al, 2014).

Pg 9 line 19 "...activity of AID unaffected.." Please add in "is unaffected"

Done.

Pg9 line 22 A reference for R-loops is required.

We have added a recent general review on R-loops.

Pg 9 line 18 and Pg 21 Methods. The authors use the DNA-RNA hybrid IP (DRIP) assay with the S9.6 antibody to assess the extent of ssDNA formation in the presence and absence of H3.3. However, recent work has shown that S9.6 Ab binds AU-rich RNA:RNA duplexes with a KD that is only 5.6-fold weaker than for RNA:DNA duplexes (Zhang et al BMC Res Notes. 2015 Apr 8;8:127. doi: 10.1186/s13104-015-1092-1). It has been reported that fold back of ssRNA can generate RNA:RNA duplexes that may bind the S9.6 antibody, and adventitious binding of RNA may produce short RNA:DNA regions (Zhang et al). Thus, use of the S9.6 antibody should be preceded by RNase A treatment to remove free ssRNA. The authors must be cautious when interpreting S9.6 data, and confirmation by independent structural and functional methods is essential.

The study by Zhang et al examines the specific case of the Ig switch regions. The authors report that they are unable to obtain an S9.6 RNA.DNA hybrid signal above background without pre treatment with RNase A. They argue this is due to double stranded RNA formed by transcripts folding back interfering with the ability of S9.6 to bind to genuine RNA.DNA hybrids. We suggest that this may be a particular issue with the highly repetitive switch regions or the precise methodology that Zhang et al employed, because it is not a problem we have with the IgV region, or that other investigators have reported when studying various non-repetitive loci. We observe a significant and reproducible enrichment with S9.6 in the variable immunoglobulin region of wild type cells that is on average 15-fold greater than the background precipitated by an irrelevant IgG (Fig 5.A in the original submission and Fig 1 below, which shows the raw data from three independent experiments).

Fig. 1 Example un-normalised raw DRIP data shown as % of input. Each group shows results from completely independent IP experiments.

As pointed out by Zhang et al, the reduction of the IP signal observed following RNaseH treatment is by itself “*indicating that the IP signals are indeed due to RNA:DNA hybrids*”. We demonstrated the specificity of our R-loop detection with S9.6 both with *in vitro* RNase H treatment and by expression of chicken RNase H1 *in vivo*.

The second, and not fully substantiated, argument made by Zhang et al. is that RNA present in the extract may form spurious RNA:DNA hybrids during the preparation of the sample. Such hybrid formation would contribute to the background of the assay. Since we show that *in vivo* expression of RNaseH1 significantly reduces R-loops in IgV of wild type cells, it is not the case that formation of RNA:DNA hybrids during the DNA extraction is a major issue in our experiments. Nonetheless, to test these assertions, we have performed a DRIP experiment at IgV in wild type and *h3.3* cells with RNase A pre-treatment (Fig. 3).

Fig. 3. DRIP at *IGVL* in wild type (WT) and *h3.3* cells with RNAse A pre-treatment.

This experiment gives us the same result as we report in the paper, namely that R-loops are significantly reduced when H3.3 is absent. It is possible that the RNAse A pre-treatment did reduce the background slightly, but the conclusion is unaltered and our existing data is clear. Thus, I do not believe that repeating all our DRIP experiments with RNAse A treatment is necessary. Importantly, we provide additional structural evidence of ssDNA in the form of our bisulphite mapping experiments.

Pg 10 line

Results using DNA-RNA hybrid IP (DRIP) with the S9.6 antibody are provided. However, there was no discussion in the text of the results regarding Figure 4A for samples that were treated with RNaseH.

We have added a sentence to discuss the RNaseH treatment in the DRIP experiments.

Pg 10 line 16 "therefore" should be deleted.

OK

Pg 11 line 4: The authors conclude that short patches of ssDNA are unaffected by RNaseH treatment and therefore are not related to the presence of R-loops since RNaseH treatment abolished R-loops but not ssDNA patches. It is not clear how one would know whether R-loops are a consequence of ssDNA by means of the RNaseH experiment they provide. It is possible that RNaseH sensitive R-loops originate from RNaseH refractory small ssDNA patches.

R-loops could well be a consequence of the formation of ssDNA, and this is precisely what we argue. R-loop formation positively correlates with the level of ssDNA, as detected by bisulphite sequencing (compare the wild type and *h3.3* cells). However, reducing R-loops at IgV in wild type cells by expressing RNase H1 in the cells does not affect the level of ssDNA or IgV diversification. This suggests that the R-loops are a consequence of the process that generates the ssDNA, not the cause of ssDNA formation.

Pg 13-14: The discussion of pausing as an alternate mechanism AID mutagenesis is somewhat misleading. RNAPII pausing is most likely a mechanism for AID recruitment through Spt5 and most likely contributes to R-loop formation and creation of ssDNA as substrate for AID. These issues are mechanistically linked. The R-loop formation may be short patch or long stretches (as in S regions). The authors' discussion of these issues needs clarity.

We have substantially revised our discussion of these points to take on board these suggestions. We do not dispute that RNAPII pausing may contribute to R-loop formation and creation of ssDNA,

which could in turn be a substrate for AID. Our data do not address the potential role for Spt5 so we cannot be sure whether the influence H3.3 exerts on the rate of diversification is downstream of Spt5 and/or a separate process that does not necessarily require RNA polymerase pausing. Our data do not provide support for a model in which H3.3 is promoting the formation of AID-accessible ssDNA through increasing RNAPII pausing, or is doing so in a subtle manner that is below the detection threshold of the DRB release assay. Further, as discussed above, our data do not provide strong support for R-loops being necessary intermediate in driving Ig diversification in this system.

Pg 15: The authors write, "The promotion of AID access by H3.3 would restrict AID activity to a relatively limited fraction of the genome and may thus contribute to preventing off-target mutagenesis. However, since H3.3 is enriched in many transcribed genes, most of which are not targeted by AID, the question still remains what is special about the Ig loci that promotes high levels of mutagenesis?" It is entirely possible that AID off-targeted genes are also enriched for H3.3, which in turn mediates formation of ssDNA patches. Do the authors know that AID off targeted genes lack H3.3? This section should be re-written.

We currently know nothing about AID off-target effects in DT40 and an exploration of this issue is beyond the scope of the current manuscript, but I think is a legitimate point for discussion in the paper. Many, if not all, transcribed genes are enriched for H3.3 and it is certainly a prediction that H3.3 would contribute to targeting of non-Ig loci. However, the point is that H3.3 is clearly not, on its own, sufficient to explain AID targeting. We are raising the possibility that other factors related to local chromatin conformation and constraints could interact with H3.3 to promote the formation of single stranded DNA, such as matrix attachment. We appreciate that these ideas are currently speculative, but feel that they present an interesting perspective on the data. We have tried to ensure that this discussion is clearly presented.

References

- Frey A, Listovsky T, Guilbaud G, Sarkies P, Sale JE (2014) Histone H3.3 is required to maintain replication fork progression after UV damage. *Curr Biol* **24**: 2195-2201.
- Taylor BJ, Nik-Zainal S, Wu YL, Stebbings LA, Raine K, Campbell PJ, Rada C, Stratton MR, Neuberger MS (2013) DNA deaminases induce break-associated mutation showers with implication of APOBEC3B and 3A in breast cancer kataegis. *Elife* **2**: e00534.
- Wang M, Yang Z, Rada C, Neuberger MS (2009) AID upmutants isolated using a high-throughput screen highlight the immunity/cancer balance limiting DNA deaminase activity. *Nat Struct Mol Biol* **16**: 769-776.

Accepted

25 April 2016

Thank you for submitting your revised manuscript for our consideration. It has now been seen once more by two of the original reviewers (see comments below), who are both satisfied with the revisions and response. I am therefore happy to inform you that we have now accepted the study for publication in The EMBO Journal.

Referee #2:

The authors had nicely addressed my concerns and comments and I feel that the manuscript represents a significant contribution to the field of somatic hypermutation.

Referee #3:

This revised manuscript is considerably improved for presentation of the data and clarity of ideas. The findings make a substantial contribution to our understand of H3.3 in generating chromatin accessibility and will be of general interest.

Corresponding Author Name: Julian E. Sale

Journal Submitted to: The EMBO Journal

Manuscript Number: EMBOJ-2016-93958